# Global inequity creates local insufficiency: A qualitative study of COVID-19 vaccine implementation challenges in low-and-middle-income countries

Victoria Haldane[1], Archchun Ariyarajah[1], Isha Berry[1], Miranda Loutet[1], Fabio Salamanca-Buentello[2], Ross E. G. Upshur[1,2]*

1 Dalla Lana School of Public Health, University of Toronto, Toronto, ON, Canada, 2 Bridgepoint Collaboratory for Research and Innovation, Lunenfeld-Tanenbaum Research Institute, Sinai Health, Toronto, ON, Canada

☯ These authors contributed equally to this work.

* ross.upshur@utoronto.ca

**Data Availability Statement:** All relevant data are within the paper and its Supporting Information files.

## Abstract

### Introduction

The COVID-19 pandemic has amplified pre-existing challenges to health promotion and care across the world, and particularly in low- and middle-income countries (LMICs). This qualitative study draws on data from a panel of immunisation experts and uses a novel framework of vaccine delivery domains to explore perspectives from those who live and work in these settings on the challenges to implementing COVID-19 vaccine programs in LMICs.

### Methods

We conducted a thematic content analysis of 96 participant free text replies to questions from Round I of a three-round Delphi consensus study amongst global experts on COVID-19 vaccine implementation.

### Results

Participant responses highlighted challenges to vaccine program implementation including issues related to equity; governance, decision-making, and financing; regulatory structures, planning, and coordination; prioritisation, demand generation, and communication; vaccine, cold chain, logistics, and infrastructure; service delivery, human resources, and supplies; and surveillance, monitoring, and evaluation.

### Conclusion

We reflect on our findings in light of global efforts to address vaccine inequity and emphasise three key areas salient to improving vaccination efforts during novel infectious disease outbreaks: 1) Ensuring safe and sustainable service delivery in communities and at points of

**Funding:** This work received funding from the University of Toronto Implementation Science Cluster Trainee Program and the University of Toronto Student Engagement Award. Funders had no role in the design and conduct of the study; collection, management, analysis, and interpretation of the data; preparation, review, or approval of the manuscript; or decision to submit the manuscript for publication.

**Competing interests:** The authors have declared that no competing interests exist.

care; 2) Strengthening systems for end-to-end delivery of vaccines, therapeutics, diagnostics, and essential supplies; 3) Transforming structural paradigms towards vaccine equity.

## Introduction

The COVID-19 pandemic has exacerbated pre-existing challenges to health promotion and care globally, and particularly in low- and middle-income countries (LMICs). Already fragile and fragmented health systems have been overwhelmed during surges in infections, human resource shortages in the health workforce have become more acute, and pandemic response measures have interrupted routine service delivery threatening global progress in areas such as malaria, tuberculosis, and HIV/AIDS [1–3]. Widespread public health and social measures, which aim to limit viral transmission and protect lives and livelihoods, have also led to unintended challenges including increasing poverty, food insecurity, and social unrest in many LMICs [4–6].

The rapid development of multiple safe and effective vaccines for emergency use in 2020 was heralded as the way out of the pandemic and towards health, social, and economic recovery. Thus, the global vaccine architecture mobilised to build, scale, and sustain national and sub-national vaccine procurement and delivery systems. This included the development of the COVID-19 Vaccines Global Access (COVAX) initiative [7]. As one of the four pillars of the Access to COVID-19 Tools Accelerator (ACT-A), COVAX is a coordinating mechanism for international resources that aims to ensure LMICs have equitable access to COVID-19 vaccines.

However, the global COVID-19 vaccination effort has thus far underscored—and indeed replicated—the pervasive inequities that shape global health [8]. The failure of multilateral initiatives, such as COVAX, to counter vaccine inequities has perpetuated the gap in health and well-being between high-income countries (HICs) and LMICs, with seemingly little political will or accountable action to equitably distribute doses. The WHO has decried not only the devastating health impacts global vaccine inequity causes, but also its "lasting and profound impact on socio-economic recovery" in LMICs [9]. Indeed, global efforts fell short of achieving multilateral targets to vaccinate 70% of the population of all countries by the middle of 2022 [10].

Against this backdrop of deepening inequity, there is little synthesised evidence from those working on COVID-19 vaccine programs in LMICs as to the specific barriers faced across countries to implement and sustain these efforts. This qualitative study draws on data from a panel of vaccine delivery experts and uses a novel framework of vaccine delivery domains to explore expert perspectives from those who live and work in these settings on challenges faced by LMICs in implementing COVID-19 vaccine programs.

## Materials and methods

We analysed our data using a modified VIRAT/VRAF 2.0 framework, which provides a comprehensive and authoritative overview of vaccine readiness comprised of a set of indicators describing key areas relating to country readiness for vaccine deployment. The framework was jointly established by the WHO and the World Bank to combine the WHO's *COVID-19 Vaccine Introduction Readiness Assessment Tool* (VIRAT) and the World Bank *Vaccine Readiness Assessment Framework* (VRAF) [11]. For the purposes of our analysis, our conceptual framework represents the building blocks of vaccine program delivery based on the VIRAT/VRAF 2.0 and grounded in equity, a key factor identified in our analysis (Fig 1).

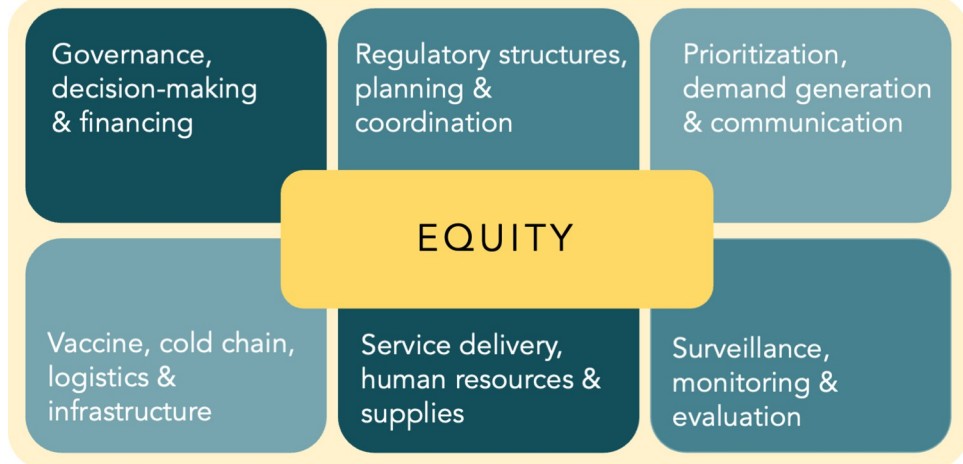

**Fig 1. Modified VIRAT/VRAF framework.**

## Study design

We conducted a three-round modified Delphi study between April and October 2021. The Delphi process is a structured and systematised method, which uses a series of sequential questionnaires interspersed with controlled feedback to collect and distil knowledge from a dispersed panel of anonymous experts to build reliable group consensus on a specific judgment issue. Delphi exercises have been used to develop consensus on priorities, methodological approaches, and definitions in immunisation research [12]. Here, we report only on data collected in Round I of the Delphi study (April-May 2021), which was designed to elicit broad and general concepts from the expert panellists using unstructured, open-ended questions. We piloted the survey with a convenience sample of researchers with global health and/or vaccine experience to gather feedback on survey length and clarity, which was incorporated into the final survey.

## Participants & recruitment

In accordance with the Delphi study design, we identified participants who were recognised experts in the field of global vaccine delivery and implementation. Individuals were eligible if they were at least 18 years of age, had access to the Internet, could read and write in English, and had been identified as experts in vaccine program implementation. We deliberately sought representativeness with respect to gender, geographic distribution, and specialty areas.

Participants were recruited using two complementary purposive and criterion sampling methods. The initial list of experts was identified through professional contacts of the study team, as well as both grey and peer-reviewed published literature. We identified expert members of publicly listed immunisation committees and panels from various global health organisations, including the Strategic Advisory Group of Experts on Immunisation (SAGE) at the World Health Organization (WHO) and the Scientific Advisory Committee of the Coalition for Epidemic Preparedness Innovations (CEPI). Participants were recruited through standardised emails that were sent to their publicly reported email addresses. Second, we utilised snowball sampling to identify additional experts who may not have been captured in the literature. Upon invitation into the study, participants were offered the opportunity to nominate up to three other experts within their network. Each new nominated participant was subsequently invited into the study, and similarly offered the opportunity to nominate additional experts.

Given that some experts may have had multiple affiliations, and therefore could have had multiple email addresses, the study team manually reviewed the list of nominated participants to remove duplicates.

## Data collection

A minimal set of socio-demographic data were collected using a structured questionnaire, including participants' self-reported gender, geographic and content areas of expertise (in which participants could select multiple options), organisational affiliation, and years of experience. Participants were then asked to identify three to five of the most important challenges in i) distribution, ii) prioritisation, and iii) administration of COVID-19 vaccines in LMICS, along with three to five potential solutions to address these challenges. To ensure consistency, we provided standardised definitions for vaccine distribution, prioritisation, and administration at the start of the survey: distribution referred to acquisition, storage, and deployment of vaccines; prioritisation referred to the order in which COVID-19 vaccines should be distributed to populations and geographical areas, and for special considerations for program implementation; administration referred to actions and mechanisms that ensure vaccines are provided to the population safely and effectively. Answers to these open-ended questions were recorded in free text boxes following each question. Data were collected using the Research Electronic Data Capture (REDCap) survey platform hosted at the University of Toronto [13]. Round I of the survey was open from April 20, 2021 to May 15, 2021; participants with incomplete responses were sent reminders on a weekly basis until survey close to increase response rate.

This study received ethical approval from the University of Toronto Research Ethics Board (Protocol Number: 40797). All participants provided online written informed consent before being permitted to access survey questions.

Patients and the public were not involved in the design, recruitment, conduct or dissemination of the study.

## Analysis

Using our guiding framework, this analysis adopted a qualitative descriptive approach, which is descriptive rather than interpretive in focus, is well suited to cross-cultural data, and has been used in other qualitative analysis of free text survey responses [14–16]. Given that our qualitative data were limited to brief statements in response to open-ended questions, the qualitative descriptive approach enabled us to organise the results without attempts to interpret our participants' meaning or "move beyond" the text as it was presented [17]. To enhance the rigour of our analysis we adopted a committee approach whereby four research team members (VH, IB, AA, ML) analysed the data in an iterative process [18]. This involved an initial meeting after each team member had read the consolidated data to discuss initial impressions and define the framework application. Then, each member analysed a subset of the data individually and assigned codes to the text using our framework. Any disagreements or differences in interpretation were resolved through discussion amongst the group.

Data were organised in QSR NVivo 12, coding was conducted first in QSR NVivo 12 and then through team discussion using Microsoft Word. Data was coded deductively using framework analysis as described by Ritchie and Lewis, while allowing for elements of thematic analysis as described by Braun and Clarke, namely inductive identification of themes [19, 20]. We then engaged in multiple rounds of discussion to iterate on the framework (e.g., adding and consolidating domains) and to refine the assignment of codes to the framework. This work

was also reviewed by our two senior authors (FSB, RU) and discussed as a team to further ensure congruence in our analytic interpretation of the data.

## Results

A total of 426 participants were invited to be involved in the Delphi study. Of these, 96 (22%) completed Round I of the survey and provided responses that were included in the content analysis. See Table 1 for participant characteristics. We organised our data using the domains identified in our conceptual framework. Table 2 offers illustrative responses mapped to our domains.

**Table 1. Baseline characteristics of participants.**

| Characteristic | N (%) |
|---|---|
| **Gender** | |
| Male | 50 (52.1%) |
| Female | 43 (44.8%) |
| Prefer not to disclose | 1 (1.0%) |
| **Region of expertise (participants could select multiple regions)** | |
| Sub-Saharan Africa | 40 (41.7%) |
| Latin America and the Caribbean | 37 (38.5%) |
| South Asia | 34 (35.4%) |
| East Asia and the Pacific | 25 (26.0%) |
| Europe and Central Asia | 23 (24.0%) |
| Middle East and North Africa | 23 (24.0%) |
| North America | 17 (17.7%) |
| Prefer not to disclose | 1 (1.0%) |
| **Organisational affiliation** | |
| Multilateral organisation (e.g., WHO, World Bank) | 36 (37.5%) |
| Research organisation or academic institution (e.g., university, college, etc.) | 24 (25.0%) |
| Government (including governmental public health organisation) | 8 (8.3%) |
| Funding/donor agency (e.g., BMGF) | 4 (4.2%) |
| Healthcare facility (e.g., hospital, clinic, nursing home, etc.) | 3 (3.1%) |
| Independent consultants | 3 (3.1%) |
| Industry (e.g., pharmaceutical company, corporation, etc.) | 3 (3.1%) |
| Public-private partnership (e.g., GAVI) | 3 (3.1%) |
| **Expertise (participants could select multiple fields)** | |
| Public health and surveillance | 61 (63.5%) |
| Policy-making and governance | 41 (42.7%) |
| Leadership and management | 35 (36.5%) |
| Program development and evaluation | 32 (33.3%) |
| Population health research | 28 (29.2%) |
| Logistics and supply chains | 23 (24.0%) |
| Vaccine development | 16 (16.7%) |
| Clinical practice | 12 (12.5%) |
| Lab-based research | 9 (9.4%) |
| Industry | 3 (3.1%) |
| **Years of expertise** | |
| 10 or more years | 76 (79.2%) |
| 5–9 years | 17 (17.7%) |
| Less than 5 years | 1 (1.0%) |

**Table 2. Illustrative responses and the building blocks of vaccine program delivery.**

| Domain | Illustrative responses |
|---|---|
| Governance, decision-making and financing | "Failure of global initiatives, such as COVAX, including the lack of transparency in their management." |
| Regulatory structures, planning and coordination | "Lack of policy and plan re deployment of vaccines or microplanning did not address all the deficient areas and identify all the required resources; Coordinating the activities of all stakeholders can be major challenges if initial planning and decision making did not include them; Not including all partners, including representatives from all geopolitical areas/regions, in planning and decision making." |
| Prioritisation, demand generation and communication | "Prioritisation based on political considerations instead of epidemiological and demographic considerations. Frequent changing of the prioritisation criteria gives mixed messages and creates mistrust." |
| Vaccine, cold chain, logistics, and infrastructure | "The shortage of COVID-19 vaccine globally, the cost of vaccines, the distribution chain, the availability of the vaccine in good quantities, the geographical factors and transport system." |
| Service delivery, human resources, and supplies | "Overcrowding and seating arrangement considering Covid-19 infection prevention; Having enough space, ventilation and volunteers, security staffs and health workers." |
| Surveillance, monitoring, and evaluation | "In many of these countries, there is no adequate surveillance system to monitor the spread of the disease, so it will be difficult to report/monitor adverse events due to immunisation." |
| (In)equity | "Unequal competition with high-income countries for the acquisition of vaccines, especially due to lack of resources, lack of knowledge of multilateral purchase mechanisms, blockades, or embargoes on some countries due to political or commercial issues, shortage of biologicals worldwide that especially affects low and middle-income countries as they are not producers or developers of vaccines" |

## Governance, decision-making & financing

Participants from across regions described how global vaccine governance structures challenge vaccine acquisition in LMICs. Similarly, participants across donor agencies, governments, and multilateral organisations alike emphasised the impact of global shortages and limited purchasing options for LMICs. One participant with expertise in South Asia emphasised that this was particularly challenging for small countries and countries with financial constraints. Another participant from a multilateral organisation elaborated on this, noting how smaller LMICs may not be in strong positions to negotiate vaccine acquisition with global suppliers. Others described how dependency on vaccine donations or initiatives such as COVAX disadvantaged LMICs. For instance, participants with expertise in East Asia and the Pacific reported that delays in receiving vaccines through COVAX were a source of failure. Another participant from a multilateral organisation reported on a lack of transparency in COVAX's management. Others emphasised how vaccine nationalism and vaccine hoarding by HICs amplified inequitable access. A participant from a multilateral organisation underscored that a lack of coordinated distribution and procurement mechanisms across countries skewed procurement towards high income and vaccine-producing countries.

At a national level, participants from all regions reported the negative influence of political interests on vaccine programs. Participants with expertise in Africa in particular, described a lack of transparent decision-making. For instance, a participant with expertise in the Middle East and North Africa identified abuse and favouritism in distribution as key challenges. Similarly, participants across multiple regions emphasised that vaccine prioritisation was influenced by corruption, with some participants including reports of 'queue jumping' by influential persons. A participant with expertise in Latin America and the Caribbean reflected

on how purchasing of vaccines at the national level posed a challenge to sub-national responses, with monopolies preventing local governments from making direct purchases from manufacturers. Another participant with expertise in South Asia described challenges relating to who makes health decisions and considered that this is often done by 'administrators' instead of health experts. As one participant from a multilateral organisation summed-up, vaccines are being used as a political tool rather than as a public health intervention.

Participants across all regions described challenges related to inadequate financial resources to procure the quantity of required vaccines, as well as inadequate and unsustainable funding for logistics and cold chain requirements. Some participants highlighted challenges related to a lack of operational funding in general, and the resultant difficulties for delivering vaccines to remote and hard-to-reach areas. A participant with expertise in Latin America also reported how a lack of domestic funding for vaccine procurement limits a country's vaccine options to only those offered through COVAX, and how this could pose challenges should there be vaccine hesitancy towards the procured vaccine.

## Regulatory structures, planning & coordination

Participants described how a lack of strong national regulatory agencies limited LMIC capacity to evaluate and approve new vaccine technologies and manufacturing processes. In addition, one participant with expertise in Latin America and South Asia noted that even once vaccines are approved, many Ministries of Health do not have the capacity and capabilities to manage programs at the required scale. Similarly, a participant with expertise in Sub-Saharan Africa reported on poor and uncoordinated implementation of vaccine programs due to overlaps in functions among tiers of government. Another participant, with expertise in East Asia and the Pacific, pointed out that different regional capacities within countries, including infrastructure, logistics, human resources, and systems, challenge planning and coordination.

Others reported inadequate implementation processes, with one participant affiliated with a donor agency describing how lack of transparent, accountable, and unbiased processes challenges vaccine distribution. Another participant, with a multilateral organisation affiliation, emphasised the challenges of national vaccine deployment capacity and preparedness in constrained health service delivery systems, with limitations in service delivery capacity slowing down vaccine uptake. A participant with expertise in Sub-Saharan Africa described how plans need not be made from scratch, underscoring challenges that stem from 'reinventing the wheel' and ignoring existing mechanisms, particularly those used for polio vaccination.

Other participants described challenges due to a lack of microplanning—a specific process in vaccine delivery used to identify priority communities, address barriers, and develop workplans with solutions [21]. For instance, one participant with expertise in Latin America emphasised challenges caused by insufficient microplanning and the need to consider all possible deficiencies and required resources. The same participant emphasised that coordinating across stakeholders can be a major challenge if the stakeholders are not included in initial planning and decision making. However, two participants reported divergent views, instead describing how recent investments in vaccine programs (such as those for Human Papilloma Virus) and already established vaccination campaigns would make distribution less of a challenge.

## Prioritisation, demand generation & communication

Participants reported numerous challenges related to prioritising the order in which different population sub-groups are permitted to access vaccines. Participants with expertise across regions reported that vaccine prioritisation should be decided by a National Immunisation

Technical Advisory Group (NITAG), with some highlighting how WHO guidance can inform prioritisation decisions. One participant with expertise in South Asia explained that while WHO guidance aims to achieve public health goals (namely to prevent overwhelming the health system and to prevent severe disease and death amongst at-risk groups), decision makers in LMICs make political prioritisation decisions based on economic objectives and political commitments. Other participants reported challenges relating to the role of bribery or favours to ensure certain groups, or certain people, are prioritised. One participant with expertise in South Asia explained how in some LMICs frequent changing of the prioritisation criteria sends mixed messages and creates mistrust amongst the public. Another participant with expertise in Latin America underscored that a major challenge is communicating to populations how priority groups were determined and why, as this is crucial to priority populations understanding when and how to access vaccines, and to ensure trust is built and/or maintained with the general population while they wait their turn. This participant went on to emphasise the importance of understanding and addressing concerns that priority populations may have regarding being the first to be selected to get vaccinated, especially in the case of health care workers.

Participants described how even with prioritisation, there may be additional challenges to demand generation and communication in LMICs, particularly related to vaccine hesitancy and misinformation. Two participants with expertise across regions highlighted challenges related to hesitancy from health workers as a priority group, and how this could have knock-on effects should health workers decline vaccination or not encourage their patients to get vaccinated. Participants across regions reported how misinformation, mistrust, social media, and inadequate communication strategies regarding vaccines can lead to hesitancy, which results in inadequate vaccine coverage. Other experts highlighted the challenges in addressing hesitancy related to the risk profiles of vaccines being used in LMICs, particularly, as one participant described, when it is perceived that HICs are using "premium vaccines" compared to LMICs. Others expanded on this and emphasised the challenges in addressing misinformation related to adverse events following immunisation (AEFIs) and how many in LMICs are taking a "wait and see" approach. One participant with expertise in Latin America and the Caribbean specifically highlighted that communication and demand generation was an area where LMICs were underinvesting and that "anything that has the potential to shake the population's trust in the vaccine, the vaccinator, or the immunisation program, can negatively impact uptake."

## Vaccine, cold chain, logistics & infrastructure

Participants across regions reported challenges related to the vaccine itself. One participant with expertise across Asia Pacific, Europe and Central Asia noted that vaccines offered through COVAX have a short expiry date, which posed logistical challenges when compounded with delays in vaccine shipment due to lengthy acquisition processes. Others from across regions reflected how unstable vaccine supplies made it challenging to ensure sufficiency and consistency, which created programmatic challenges for making decisions on timing of administration for vaccine types requiring second doses. Participants from across regions reported challenges in LMICs regarding vaccines requiring cold chain. For example, one participant with expertise in North America highlighted that mRNA vaccines, in particular the Pfizer-BioNTech vaccine, may not be appropriate candidates for remote communities due to inadequate cold chain capacities. Others from across regions reported how many LMICs not only have limited cold chain capacity but fragile supply chains in general, and widely differing capacities within countries.

Some participants related these supply chain issues to larger interconnected logistics challenges to vaccine distribution in LMICs. Participants from across regions described logistical challenges including complexities of distributing multiple types of COVID-19 vaccines at the same time, managing expiry of products with short delivery timelines and/or narrow temperature storage ranges, issues to mitigate theft or falsification of COVID-19 vaccines in the primary supply chain or adjacent to it, as well as transport issues such as fuel shortages, difficulties in obtaining transport permits, and road conditions and safety. One participant with expertise in Sub-Saharan Africa reflected on inadequate infrastructure including unreliable power grids to maintain cold chain integrity, while a participant with expertise in Latin America highlighted inadequate facilities to serve as vaccination centres, or centres not placed in locations acceptable or accessible to the population. Others from across regions emphasised challenges brought about by inadequate mobile and outreach activities, and the significant costs associated with creating accessible, acceptable, and adequate infrastructure to safely delivery large-scale COVID-19 vaccination programs.

## Service delivery and human resources & supplies

Many participants from across regions reported challenges in service delivery related to reaching remote populations in LMICs. One participant with expertise in North America highlighted a lack of safe roads or airports for delivery of supplies that enable service provision. Others from across regions reported that geographical limitations will ultimately bias prioritisation with remote areas being excluded. One participant affiliated with a multilateral organisation reflected on previous experience with polio vaccination campaigns in Afghanistan and Pakistan in which door-to-door vaccine campaigns were successful to overcome challenges with populations reaching vaccination centres. Participants across regions reported that areas with armed conflicts would also be unreachable for vaccine campaigns. One participant with expertise in the Middle East and North Africa underscored the challenges in vaccinating populations along borders and how ensuring service delivery in these regions requires negotiating agreements between governments.

When considering vaccine service delivery, some participants called into question the capacity and preparedness of LMICs given their already constrained health systems. Others highlighted that few LMICs have adult immunisation programs across the life course in place, thus programmatic readiness for mass adult immunisation was raised as a significant challenge. One participant with expertise in South Asia noted that vaccine programs in LMICs would be further challenged should they need to administer multiple COVID-19 vaccine types at the same time.

Human resource challenges were reported by many participants across regions as a key barrier to vaccination efforts. Participants described an insufficient number and inadequate capacity of vaccinators and a need to train health workers not only for administering vaccines but also for screening and post-vaccine observation. Participants with expertise across regions described how shortages of workers were related to routine immunisation workers being deployed to provide care as part of the COVID-19 response. Participants' responses across regions also emphasised that a lack of personal protective equipment (PPE), needles, syringes, cold packs, and other vaccine delivery materials posed an ongoing challenge to ensure sustainable and safe vaccine service delivery. As one participant with expertise in South Asia summarised: ensuring enough space, ventilation, volunteers, security staff, and health workers was a challenge for infection prevention and control in the context of overcrowded service delivery sites.

### Surveillance and monitoring & evaluation

Participants across regions reported challenges related to poor pharmacovigilance systems, which are of great importance given the administration of new vaccines. Participants also reported the importance of having adequate surveillance capacity to identify and provide timely management of AEFIs. One participant from the Middle East and North Africa region reported how inadequate surveillance systems in general made it difficult to monitor the progress and effectiveness of immunisation activities. Another participant with experience across regions highlighted how limited systems for tracking and follow-up were a challenge to ensuring that two-dose vaccines were appropriately scheduled and provided. Others described challenges to monitoring and evaluation related to inadequate systems infrastructure. For example, monitoring systems were also limited for tracking and monitoring the distribution of vaccines and other key supplies, with disaggregated information systems leading to incomplete or inaccurate data about vaccine target populations.

### (In)equity

Our analysis underscores the inequities in global vaccine efforts and the way these, along with the pervasive mistrust that characterises the COVID-19 infodemic, shape vaccine programs. Participants described global inequities in distribution, access, and availability of vaccines, with most emphasising those between HICs and LMICs. Two participants with expertise in Latin America and the Caribbean described how "elite capture" is the biggest challenge. One participant with experience across regions explained how the limited supply to LMICs created a sense of being "left out." Another participant emphasised that economic inequities underpinning vaccine acquisition are a result of limited progress on decolonisation. Others highlighted inequities within countries where unequal geographic access to health services and distribution infrastructure deepen the urban-rural health divide. One participant affiliated with a multilateral organisation highlighted how migrants are particularly vulnerable to exclusion due to their lack of access to health services. Another participant with expertise in Latin America linked these inequities to public policy, explaining that inequities will be deepened if decision makers don't take into consideration the fragility and vulnerability of some communities, in particular Indigenous communities. Another described the consequences of inequity as increased corruption and the formation of an 'underground market' for vaccines. Participants also reported a lack of trust challenging all levels of the COVID-19 vaccination effort.

## Discussion

COVID-19 vaccine program implementation has challenged health systems globally, with a disproportionate and inequitable impact on LMICs. In this qualitative content analysis, we have used a novel framework adapted from the VRAT/VRAF to elaborate on challenges from the perspective of experts with direct experience in vaccine program implementation in LMICs. In doing so, we highlight not only challenges across key domains, but also emphasise the importance, and current deficit, of equity in vaccine program implementation. While we report and reflect on challenges in the context of LMICs, we note that many of the identified challenges and solutions have been faced by HICs as well [22]. Together, our findings have noteworthy implications for supporting pandemic recovery.

With an aim towards global vaccine equity, several multilateral groups including the Independent Panel for Pandemic Preparedness and Response (IPPPR) and the Pan-European Commission on Health and Sustainable Development, amongst others, have made recommendations for immediate actions to be taken to ensure doses reach LMICs [23, 24]. The Director General of WHO also has repeatedly called global vaccine inequity a 'solvable problem' that

continues to undermine efforts to end the pandemic [25–27]. Our findings underscore the global nature of vaccine inequity, as participants from across regions reflected on similar challenges to vaccinating their populations including the need to strengthen local implementation and national operational capacity, as well as structural drivers of ongoing inequity.

Ending the current pandemic and preventing future outbreaks ultimately relies on local action. Whether it is detecting—or responding to—outbreaks, communicating public health and social measures, or mobilising vaccine campaigns, supporting front line workers to deliver care safely and effectively is critical to programmatic success. Our results highlight the challenges faced at the front lines in safely implementing vaccination programs during a public health emergency, across a range of settings, and amongst competing demands and differing levels of community trust [28]. Beyond the need for ensuring sufficient cold chain capacity to deliver vaccines, participants identified other challenges related to unmet development goals and longstanding inequities including interrupted power supply, challenging road conditions, and fuel shortages. While our findings do point to the threat of misinformation and disinformation, they also highlight how lack of trust and transparency in decision-making during the pandemic threatened vaccine uptake and acceptance. Front line health workers are facing these challenges directly in communities. In addition to persistent human resource shortages in the health sector in LMICs, our findings show that these workers may be insufficiently trained, and inadequately protected. Without global and national investment in training and protection, health workers globally will be unable to safely and sustainably carry out vaccination campaigns at the scale and for the duration required [29].

Our findings also emphasise the importance of ensuring that LMICs not only have agency to effectively procure consistent and sustainable supplies of vaccines, but also the necessary infrastructure and supplies to ensure uninterrupted and safe delivery [30]. ACT-A is an important global coordinated effort to accelerate development, production, and equitable access to COVID-19 tests, treatments, and vaccines [31]. Our results highlight the need to extend this platform to consider the entire suite of vaccine delivery needs. These include working with countries to support program delivery, transforming access to the vaccine product procurement, addressing supply chain gaps, and ensuring that front line needs are met at the point of care.

While our findings emphasise upstream global and structural issues, they also offer actions that national governments must take to better prepare for vaccine procurement and service delivery from global mechanisms. Strengthening regulatory agencies and processes is crucial to ensuring swift approval of new vaccines and technologies, particularly during public health emergencies [32]. In addition, our findings highlight the importance of building on existing platforms such as vaccination plans established for polio or other mass vaccination campaigns to facilitate service delivery strengthening [33]. Health systems strengthening before emergencies includes investing in surveillance systems from the bottom up not only to monitor coverage, but to ensure that those at the front lines can detect, report, and gather national data on AEFI emergencies [34]. Concerted efforts are needed to strengthen health systems so LMICs are better prepared to receive and deploy vaccines, therapeutics, diagnostics and essential supplies, and to link them with monitoring and delivery systems during emergencies, while also ensuring ongoing routine health services [35].

Our findings also emphasise the challenges faced by LMICs in procuring and receiving doses compared to HICs. These challenges are even more pressing as HICs administer booster doses despite limited vaccine coverage globally [36]. Our participants identify HICs as hoarding vaccines and supplies, as well as having an unfair advantage in vaccine acquisition. This has been seen throughout the pandemic, beginning with early shortages of PPE, supplies and medicines, and continuing largely unabated across waves of infection. Indeed, little has been

done to address the structural drivers that ensure HICs wield neo-colonial negotiating power over LMICs in times of collective crisis [37].

These structural drivers, which importantly include the commodification of global public goods and the dominance of neoliberalism in global health delivery, are not new and have driven many of the persisting health crises faced globally before COVID-19 [38, 39]. The long-standing view of vaccines and essential medications as a market commodity rather than a public good has been framed as not only a moral issue but also a matter of global biosecurity and pandemic preparedness [40]. This is reflected in our findings where uncertain vaccine supplies, dictated by opaque global mechanisms, and provided by countries prioritising their own interests, determine national vaccine programs in LMICs. Uncertainty ultimately leaves LMICs beholden to donors, and unable to plan and prepare for vaccine program implementation as their populations remain vulnerable to a novel infectious threat. Further, inequities and the paradigms that uphold them may be replicated within regions and countries, as seen through many respondents discussing political interests rather than scientific evidence or ethical principles as determinants of prioritisation of vaccine access. Thus, efforts to dismantle global inequities must be matched by efforts to dismantle all inequity. Without transformational change, global inequity will ensure local insufficiency to meet vaccination goals, thus threatening lives and livelihoods globally now and in years to come.

## Strengths and limitations

Our study used an existing framework to situate the findings and drew on a wide breadth of expert perspectives to inform our analysis. This enabled us to highlight the constellation of challenges and potential solutions to LMIC vaccine implementation. A further strength is the immediate relevance of the analysis to current discourse driving future pandemic preparedness efforts including recommendations from the IPPPR and other groups. On the other hand, the fact that our respondents were able to select multiple regions of expertise could be a limitation, as we were unable to robustly report on regional differences in implementation challenges. However, given that our participants were informed that our Delphi was a global consensus building exercise, their responses were overall more generalised.

## Conclusion

There are multiple and multifactorial challenges to COVID-19 vaccine program implementation in LMICs including issues of governance, decision-making, and financing; regulatory structures, planning, and coordination; prioritisation, demand generation, and communication; vaccine, cold chain, logistics, and infrastructure; service delivery, human resources, and supplies; as well as surveillance, monitoring, and evaluation. The multiple domains necessary for successful vaccine program implementation are impacted by deficits in equity and trust that disproportionately impact those living in LMICs. To strengthen equitable vaccine implementation in LMICs requires coordinated and comprehensive global efforts across domains. Importantly, concerted efforts must be made to tackle global health inequities and foster trust in institutions, decision makers, and health programs before, during, and after crises, particularly in light of the imminence of future public health emergencies.

## Supporting information

**S1 Checklist. COREQ (COnsolidated criteria for REporting Qualitative research) checklist.**
(PDF)

**S1 File.**
(XLS)

## Acknowledgments

The authors thank the study panelists for their contributions to this study.

## Author Contributions

**Conceptualization:** Victoria Haldane, Archchun Ariyarajah, Isha Berry, Miranda Loutet.

**Data curation:** Victoria Haldane, Archchun Ariyarajah, Isha Berry, Miranda Loutet.

**Formal analysis:** Victoria Haldane, Archchun Ariyarajah, Isha Berry, Miranda Loutet.

**Funding acquisition:** Victoria Haldane, Archchun Ariyarajah, Isha Berry, Miranda Loutet.

**Investigation:** Victoria Haldane, Isha Berry.

**Methodology:** Victoria Haldane.

**Project administration:** Victoria Haldane, Isha Berry, Miranda Loutet.

**Supervision:** Fabio Salamanca-Buentello, Ross E. G. Upshur.

**Validation:** Archchun Ariyarajah, Isha Berry, Miranda Loutet, Fabio Salamanca-Buentello, Ross E. G. Upshur.

**Visualization:** Victoria Haldane.

**Writing – original draft:** Victoria Haldane, Archchun Ariyarajah, Isha Berry, Miranda Loutet.

**Writing – review & editing:** Victoria Haldane, Archchun Ariyarajah, Isha Berry, Miranda Loutet, Fabio Salamanca-Buentello, Ross E. G. Upshur.

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
