## [Decision Letter · Decision Letter 0]

22 Nov 2022

PONE-D-22-15264Global inequity creates local insufficiency: A qualitative study of COVID-19 vaccine implementation challenges in low-and-middle-income countries.PLOS ONE

Dear Dr. Upshur,

Thank you for submitting your manuscript to PLOS ONE. After careful consideration, we feel that it has merit but does not fully meet PLOS ONE’s publication criteria as it currently stands. Therefore, we invite you to submit a revised version of the manuscript that addresses the points raised during the review process.

ACADEMIC EDITOR:  1. Go through reviewer 1 comments and address each of them 2. Explain data availability for this manuscript3. We need to understand if you have already published or have under review multiple articles from the same dataset, and if so, you must specify reasons as to why that data needs to be published separately from this article. ==============================

Please submit your revised manuscript by 06 January 2023. If you will need more time than this to complete your revisions, please reply to this message or contact the journal office at plosone@plos.org. Please include the following items when submitting your revised manuscript:A rebuttal letter that responds to each point raised by the academic editor and reviewer(s). You should upload this letter as a separate file labeled 'Response to Reviewers'.A marked-up copy of your manuscript that highlights changes made to the original version. You should upload this as a separate file labeled 'Revised Manuscript with Track Changes'.An unmarked version of your revised paper without tracked changes. You should upload this as a separate file labeled 'Manuscript'.If applicable, we recommend that you deposit your laboratory protocols in protocols.io to enhance the reproducibility of your results. Protocols.io assigns your protocol its own identifier (DOI) so that it can be cited independently in the future. For instructions see: https://journals.plos.org/plosone/s/submission-guidelines#loc-laboratory-protocols. Additionally, PLOS ONE offers an option for publishing peer-reviewed Lab Protocol articles, which describe protocols hosted on protocols.io. Read more information on sharing protocols at https://plos.org/protocols?utm_medium=editorial-email&utm_source=authorletters&utm_campaign=protocols.

We look forward to receiving your revised manuscript.

Kind regards,

Saurav Basu, M.D.

Academic Editor

PLOS ONE

Additional Editor Comments:

You are requested to make the changes as per reviewer comments especially with regards to data availability. Also any publications (published or under review) associated with the same dataset need to be clarified with reasons

Reviewers' comments:

Reviewer's Responses to Questions

**Comments to the Author**

1. Is the manuscript technically sound, and do the data support the conclusions?

Reviewer #1: Yes

Reviewer #2: Yes

2. Has the statistical analysis been performed appropriately and rigorously? 

Reviewer #1: N/A

Reviewer #2: Yes

3. Have the authors made all data underlying the findings in their manuscript fully available?

Reviewer #1: No

Reviewer #2: Yes

4. Is the manuscript presented in an intelligible fashion and written in standard English?

Reviewer #1: Yes

Reviewer #2: Yes

5. Review Comments to the Author

Reviewer #1: The study as a whole was well-designed and developed. I have only a few minor edits to recommend.

1) The data should be made fully available or explicitly justify not making the data available.

2) There are a few minor typos, e.g., page 4 that says, "Figure 1 about here" that should be corrected. The grammar is solid overall, but would benefit from a final review.

3) The authors should explain why they have broken their data out into multiple publications, including only the Round 1 data here. It would help to have all of the rounds in context here or justify why the other rounds are not included.

4) Please identify the countries represented in the study for context (even a range of how many countries are represented would be helpful).

5) Please list the response rate to invitations to participate in the study.

Reviewer #2: This manuscript is well written. I only have concerns for some 'over-looked' absurdities. For instance, some sections of the tables that bear the results of the study still retain the flavour of the survey instrument, e.g. 'select all that apply'. You should read through the entire paper and eliminate typographical issues from it. Overall, this is a very good paper on the subject matter. It has established that global inequity is a driver of COVID-19 vaccine programme implementation insufficiency in LMICs. I am sure that the paper, when published, will be a good addition to the efforts that are being directed at recovery from the pandemic.

6. PLOS authors have the option to publish the peer review history of their article (what does this mean?). If published, this will include your full peer review and any attached files.

Reviewer #1: No

Reviewer #2: No

---

## [Author Response · Author response to Decision Letter 0]

12 Jan 2023

Journal Requirements:

Comment 1. Please ensure that your manuscript meets PLOS ONE's style requirements, including those for file naming. The PLOS ONE style templates can be found at

Response 1. We have updated our manuscript to adhere to PLOS ONE’s style requirements

Comment 2. In your Data Availability statement, you have not specified where the minimal data set underlying the results described in your manuscript can be found. PLOS defines a study's minimal data set as the underlying data used to reach the conclusions drawn in the manuscript and any additional data required to replicate the reported study findings in their entirety. All PLOS journals require that the minimal data set be made fully available. For more information about our data policy, please see http://journals.plos.org/plosone/s/data-availability.

Response 2: We have provided a de-identified version of the data set. It is available here:

Haldane, V; Ariyarajah, Archchun; Berry, Isha; Loutet, Miranda; SALAMANCA-BUENTELLO, FABIO; Upshur, Ross EG (2023): Round 1 Delphi responses (de-identified) - COVID-19 vaccine consensus study (LMICs). figshare. Dataset. https://doi.org/10.6084/m9.figshare.21825135.v1

Comment 3. Please review your reference list to ensure that it is complete and correct. If you have cited papers that have been retracted, please include the rationale for doing so in the manuscript text or remove these references and replace them with relevant current references. Any changes to the reference list should be mentioned in the rebuttal letter that accompanies your revised manuscript. If you need to cite a retracted article, indicate the article’s retracted status in the References list and also include a citation and full reference for the retraction notice.

Response 3. We have reviewed the reference list and all is in order.

Comment 4. Also, any publications (published or under review) associated with the same dataset need to be clarified with reasons.

Response 4. The complete results of the Delphi have previously been published as:

Ariyarajah A, Berry I, Haldane V, Loutet M, Salamanca-Buentello F, Upshur RE. Identifying priority challenges and solutions for COVID-19 vaccine delivery in low-and middle-income countries: A modified Delphi study. PLOS Global Public Health. 2022 Sep 8;2(9):e0000844. Available at: https://journals.plos.org/globalpublichealth/article?id=10.1371/journal.pgph.0000844#references

The manuscript currently under review is a companion to this Delphi. Whereas the full Delphi process leads to consensus, in the initial phase of the study many participant perspectives were collected that offer a breadth of important considerations for vaccine implementation in low- and middle-income settings. This study uses qualitative methods to explore these perspectives and in doing so differs significantly from the Delphi and sheds different light on COVID-19 vaccine program implementation.

Reviewer #1: 

Comment 5. The data should be made fully available or explicitly justify not making the data available.

Response 5. TBD. 

Comment 6. There are a few minor typos, e.g., page 4 that says, "Figure 1 about here" that should be corrected. The grammar is solid overall but would benefit from a final review.

Response 6. Thank you, we have copy edited the document. We have removed the placeholder text for figure 1. 

Comment 7. The authors should explain why they have broken their data out into multiple publications, including only the Round 1 data here. It would help to have all of the rounds in context here or justify why the other rounds are not included.

Response 7. We have chosen to focus on our Round 1 data in this manuscript as it offers a breadth of perspectives on COVID-19 vaccine program implementation. The goal of the Delphi is to generate consensus, which while important, by nature distills and ranks many challenges and solutions into only a few. It is important that the diverse perspectives offered in the free text responses of the first phase have a place to be explored and discussed, which is not within the scope or aims of Delphi, but which is well suited to qualitative analysis as presented here. All rounds of data are not included here as only the first round involves gathering qualitative data – subsequent rounds use quantitative methods to determine agreement and ranking, thus making these data inappropriate for content analysis. 

Comment 8. Please identify the countries represented in the study for context (even a range of how many countries are represented would be helpful).

Response 8. We have reported on the region of expertise of the panel that answered the survey questions in Table 1. We asked in this way as many participants in our target population have expertise across several countries, as well as challenges in the distinction between an expert’s country of origin, country of residence, or country where they work. Phrasing it as country of expertise and allowing for multiple selections allows our participants to self-identify.

Comment 9. Please list the response rate to invitations to participate in the study.

Response 9. Thank you for bringing this to our attention. We have now added the response rate in the results section, line 174. The section now reads “A total of 426 participants were invited to be involved in the Delphi study. Of these, 96 (22%) completed Round I of the survey and provided responses that were included in the content analysis.” 

Reviewer #2: 

Comment 10. This manuscript is well written. I only have concerns for some 'over-looked' absurdities. 

For instance, some sections of the tables that bear the results of the study still retain the flavour of the survey instrument, e.g. 'select all that apply'. You should read through the entire paper and eliminate typographical issues from it. 

Overall, this is a very good paper on the subject matter. It has established that global inequity is a driver of COVID-19 vaccine programme implementation insufficiency in LMICs. I am sure that the paper, when published, will be a good addition to the efforts that are being directed at recovery from the pandemic.

Response 10. Thank you for your support of the manuscript. We have clarified in table 1 that the ‘select all that apply’ is an indication that participants could select as many regions as they felt were their region of expertise. The title of the category now says, “Region of expertise (participants could select multiple regions).” We hope this clarifies and adequately addresses the reviewer’s concern.

---

## [Editor Report · Decision Letter 1]

23 Jan 2023

Global inequity creates local insufficiency: A qualitative study of COVID-19 vaccine implementation challenges in low-and-middle-income countries.

PONE-D-22-15264R1

Dear Dr. Ross Upshur

We’re pleased to inform you that your manuscript has been judged scientifically suitable for publication and will be formally accepted for publication once it meets all outstanding technical requirements.

Kind regards,

Saurav Basu, M.D.

Academic Editor

PLOS ONE
---

## [Editor Report · Acceptance letter]

3 Feb 2023

PONE-D-22-15264R1 

Global inequity creates local insufficiency: A qualitative study of COVID-19 vaccine implementation challenges in low-and-middle-income countries. 

Dear Dr. Upshur:

I'm pleased to inform you that your manuscript has been deemed suitable for publication in PLOS ONE. Congratulations! Your manuscript is now with our production department. 

Kind regards, 

on behalf of

Dr. Saurav Basu 

Academic Editor

PLOS ONE